# Assessment of the Impact of Dietary Supplementation with Epigallocatechin Gallate (EGCG) on Antioxidant Status, Immune Response, and Intestinal Microbiota in Post-Weaning Rabbits

**DOI:** 10.3390/ani14203011

**Published:** 2024-10-18

**Authors:** Dafei Yin, Zhan Zhang, Yanli Zhu, Ze Xu, Wanqin Liu, Kai Liang, Fangfang Li

**Affiliations:** College of Animal Science and Veterinary Medicine, Shenyang Agricultural University, Shenyang 110866, China; yindafei@syau.edu.cn (D.Y.);

**Keywords:** weaning stress, antioxidant capacity, microbiota, rabbits, EGCG

## Abstract

Investigating strategies to mitigate weaning stress in rabbits holds considerable relevance for the rabbit breeding industry. Epigallocatechin gallate (EGCG), the predominant polyphenol in green tea, is recognized for its extensive health benefits. This study aimed to investigate the impact of dietary supplementation with EGCG on the antioxidant status, immune response, and intestinal microbiota in post-weaning rabbits. The results indicated that EGCG could effectively alleviate oxidative stress induced by weaning, as evidenced by enhanced antioxidant capacity, reduced peroxide generation, increased antioxidant enzymatic activity, improved intestinal morphology, decreased intestinal inflammation, and modulated intestinal microbiota flora. This information is essential for rabbit breeders seeking effective methods to mitigate post-weaning stress.

## 1. Introduction

In response to escalating economic and societal pressures, early weaning has emerged as a prevalent practice within the rabbit breeding industry. In semi-intensive systems, bunnies are usually weaned at 35–40 days [1]. During the initial 10–15 days post-weaning, the sudden transition increases vulnerability to gastrointestinal disorders, including weaning diarrhea and microbial disease, disrupting intestinal morphology and physiological function, and elevating mortality rates, ultimately leading to economic losses [2,3]. Additionally, early weaning compromises the welfare of bunnies by imposing considerable stress, which adversely affects body weight gain.

Weaning stress in young mammals is closely associated with oxidative stress and apoptosis, which can be attributed to various dietary and environmental factors [4]. Oxidative stress, driven by reactive oxygen species (ROS), reflects an imbalance between the generation of free radicals and the antioxidant capacity of animals [5]. Research suggests that oxidative stress in mammals can compromise intestinal barrier integrity and weaken the immune system [6]. In response to oxidative stress, endogenous antioxidant enzymes, like superoxide dismutase and catalase, exhibited increased activity in rabbits, working to mitigate oxidative damage [7]. Furthermore, the gut microbiota plays a crucial role in host health by contributing to inflammation suppression [8]. Several studies have investigated strategies to alleviate oxidative stress induced by weaning, with preliminary evidence suggesting that tea polyphenols offer potent therapeutic benefits by enhancing the antioxidant status in mammals [9].

Previous studies have demonstrated the antimicrobial, antioxidative, and antiviral properties of polyphenols [10]. EGCG, the predominant polyphenol in green tea, is recognized for its numerous health benefits, largely attributed to its phenolic hydroxyl groups [11,12]. Saffari and Sadrzadeh [13] reported that EGCG is a powerful antioxidant, protecting cell membranes from oxidative stress. Deng et al. [14] found that tea polyphenols possess immunomodulatory potential in weaning piglets experiencing oxidative stress. However, the bioavailability of EGCG varies depending on the dosage and the physiological status of the subject. For instance, a diet containing 1% EGCG was effective in reducing the inflammatory response in a mouse model [15], while the supplemental levels of 300, 400, and 600 mg/kg EGCG mitigated oxidative damage in poultry [16,17]. Nonetheless, the effects of varying EGCG dosages on post-weaning rabbits remain uncertain.

Research on the effects and mechanisms of EGCG in alleviating oxidative stress in weaning rabbits remains limited. We hypothesized that EGCG supplementation in rabbit diets could cause them to exhibit similar regulatory functions as observed in other species. The current study aimed to examine the protective effect of EGCG on antioxidant status, immune function, intestinal morphology, and microbiota composition in post-weaning rabbits. Thus, this study is expected to provide valuable insights into the potential of EGCG as an effective strategy for alleviating weaning stress in rabbits.

## 2. Materials and Methods

### 2.1. Materials

The EGCG (purity ≥ 98%) used in this study was purchased from Xi’an Tongze Biotech Co., Ltd. (Xi’an, China). It was extracted from green tea and presented as a brown power. The EGCG was food-grade.

### 2.2. Experimental Design and Animal Treatment

A total of 144 Ira rabbits (both male and female) with an average weight of 500 g, approximately 5 weeks old in good clinical health, were obtained from Shenyang Hongyu Animal Sales Co., Ltd. (Shenyang, China). The rabbits were housed in a controlled environment of 24 °C with ad libitum access to feed and water in 16 h light and 8 h dark cycle conditions at the Livestock Unit of Shenyang Agricultural University. The animals were maintained in accordance with the protocols approved by the Shenyang Agricultural University Laboratory Animal Welfare and Ethical Center (IACUC Issue No.: 22031204).

The rabbits were randomly assigned to six treatment groups, each comprising six replicates with four rabbits (both female and male) per replicate. Males and females were housed in separate cages. The experimental groups included one group without EGCG and five groups supplemented with EGCG at concentrations of 200, 400, 600, 800, and 1000 mg/kg. The control group received a basal diet, while the EGCG groups received a basal diet mixed with the respective dosses. All ingredients and additives were mixed simultaneously and then pelleted. The basal diet met NRC allowances [18], consisting of a concentrated feed mixture mixed with peanut vine as roughage. The ingredients and nutrient composition in the basic diet are detailed in Table 1. The overall experimental duration was 55 days, comprising a 7-day pre-feeding period followed by a 48-day formal experimental phase.

### 2.3. Growth Performance Traits

The rabbits were weighted one by one at 40, 68, and 88 days old. The weighing was carried out in the early morning before the animals ingested any feed or water. The leftovers were weighed and subtracted from the total amount provided to calculate the feed intake. The average daily weight gain (ADG), average daily feed intake (ADFI), and feed conversion ratio (FCR) for each rabbit were calculated.

### 2.4. Sampling and Collection

On day 87, 6 randomly chosen rabbits in each treatment group (one rabbit per replicate, 3 males and 3 females in each treatment) were selected to collect blood samples (3 mL) from the marginal ear vein. The samples were centrifuged at 3000 rpm for 20 min to obtain serum, which was then stored at −20 °C until testing the biochemical indexes. On day 88, an average weight of the rabbits from each replicate was selected, they were weighed, and then they were sacrificed at the slaughterhouse of Shenyang Agricultural University (3 males and 3 females in each treatment). Sticking was conducted immediately after stunning and the carcass was bled for a few minutes. The abdomen was opened, and the small intestine was carefully taken out. Segments of the duodenum, jejunum, and ileum were collected from each rabbit and quickly washed with saline water. The tissue samples were then fixed in 4% formalin solution for histopathological measurements. The jejunum was then opened carefully along its length and rinsed gently with a cold PBS solution. The scraped jejunal mucosa samples were frozen in liquid nitrogen and then quickly stored at −80 °C for further analysis. Approximately 2 g of apex liver from the left side were taken and stored at −80 °C for testing using the antioxidant-related index. The immune organs such as the spleen, thymus, and sacculus rotundus were dissected and weighed. The immune organ index was found using the following simple formula: weight of the immune organ (g)/body weight (g). Caecum contents were immediately collected and frozen in liquid nitrogen and stored at −80 °C for microbiota analysis. The experimental waste, including rabbit carcasses, was disposed of by burial. The entire slaughter and dissection process followed the protocols approved by the Shenyang Agricultural University Laboratory Animal Welfare and Ethical Center (IACUC Issue No.: 22031204).

### 2.5. Indicator Measurements

#### 2.5.1. Antioxidant Indicators

Liver tissue and intestinal mucosa were weighed with a precision of ±0.1 g. Then, a mixture was made by adding nine times the amount of saline in an ice water bath, creating a 10% tissue homogenate. The prepared homogenate was centrifuged at 3000× *g* for 10 min, and the liquid on top was gathered. The total antioxidant capacity (T-AOC), glutathione peroxidase (GSH-Px), hydrogen peroxide (H_2_O_2_), superoxide dismutase (SOD), and malondialdehyde (MDA) were measured using assay kits from Beijing Sino-UK Institute of Biological Technology (Beijing, China; Kit Nos.: T-AOC:HY-60021; GSH-Px: HY-M0004; H_2_O_2_:HY-M0019; SOD:HY-M0001; MDA:HY-M0003) with a spectrophotometer (Shimadzu UV-1201, Kyoto, Japan).

#### 2.5.2. Serum Immune Levels

Quantification of the immunoglobulin G (IgG), immunoglobulin A (IgA) and immunoglobulin M (IgM) in the serum samples was assayed using ELISA kits according to the protocols provided by the manufacturer (Nanjing Jiancheng Bioengineering Institute, Nanjing, China; Kit Nos.: IgG:H106-1-1; IgA:H108-1-2; IgM:H109-1-1).

#### 2.5.3. Intestinal Morphology Analysis

The formalin-fixed intestinal samples were embedded by paraffin. The detailed process was the following: a 0.5 cm sample was cut from the middle of each ileal section and then dehydrated with ethanol, cleared with xylene, and placed into embedding wax. Tissue sections (5 μm) were stained using hematoxylin (Sigma, Shanghai, China) and eosin (Sigma, Shanghai, China) and measured for histomorphology. The villous height (VH) and crypt depth (CD) were measured from 6 randomly selected villi and associated crypts [19]. The ratio of VH to CD was calculated. The sections were observed using an optical microscope (Nikon Ni-U, Tokyo, Japan) using ProgRes CapturePro software, version 2.7 (Jenoptik, Jena, Germany), at 4 × 50 magnification.

### 2.6. 16 s rRNA Sequencing and Data Analysis

The DNA extraction, library preparation, and sequencing analyses were performed by the Novogene Company (Beijing, China).

#### 2.6.1. DNA Extraction and Sequencing of 16s rRNA Gene

The total bacterial DNA in the cecal digesta samples from 36 rabbits was extracted by the CTAB method, ending with 1 ng/μL purified DNA solution. The V3–V4 regions of the bacterial 16S rRNA gene were amplified with the specific primers 341F (5′-CCTACGGGNGGCWGCAG) and 805R (5′-GACTACHVGGGTATCTAATCC). All PCR mixtures contained 15 μL of Phusion^®^ High-Fidelity PCR Master Mix (New England Biolabs, Beijing, China), 0.2 μM of each primer, and 10 ng target DNA. The cycling conditions consisted of an initial denaturation at 98 °C for 1 min, followed by 30 cycles at 98 °C (10 s), 50 °C (30 s), and 72 °C (30 s), and a final 5 min extension at 72 °C.

#### 2.6.2. Processing of Sequencing Data

Sequencing libraries were generated with a NEB Next^®^ Ultra ^TM^ IIDNA Library Prep Kit (New England Biolabs, MA, USA, Catalog: E7645) following the manufacturer’s recommendations. Finally, 250 bp parried-end reads were generated. Based on the unique barcodes of the samples, these reads were assigned to samples and truncated by cutting off the barcodes and primer sequences, and then they were merged using a fast and accurate analytical tool named FLASH (V1.2.1 1). Quality filtering on the raw tags was performed using the Fastp (V 0.23.1) software to obtain high-quality clean tags and compare them with the reference database (Silva database https://www.arbsilva.de/ for 16S, accessed on 30 October 2022) to obtain the effective tags. Denoising was performed using QIIME2 (V 2-202006) software to obtain initial Amplicon Sequence Variants (ASVs), and then the ASVs with less than 5 abundances were filtered out. QIIME2 software was also used to annotate species, and the annotation database used was the Silva database.

#### 2.6.3. Bioinformatics Analysis

Alpha diversity was calculated using the following 5 indexes: Chao1, dominance, Shannon index, Simpson index, and Pielou’s evenness index (Pielou_e). Beta diversity was calculated based on weighted UniFrac distances. Principal Coordinate Analysis (PCoA) and Non-Metric Multi-Dimensional Scaling (NMDS) were performed to obtain visual differences between groups. Adonis function analysis was used to study the significance differences.

To find out the significantly different species at each taxonomic level, the LEfSe software (V 1.0) was used to perform LEfSe analysis (LDA score > 2.0) so as to identify the biomarkers. Spearman’s analysis was conducted to analyze the potential association between the microbiota flora and antioxidant capacity based on the relative abundances at ASV levels. Further, PICRUSt2 (V 2.1.2-b) was used to predict the potential KEGG functions of the communities in the samples.

### 2.7. Statistical Analysis

All data underwent one-way ANOVA. Orthogonal polynomial contrast was used to determine the linear and quadratic effects of SPSS statistical software (v 20.0; Chicago, IL, USA). Probability values below 0.05 were considered to be significant.

## 3. Results

### 3.1. Growth Performance

Table 2 displays the data for the body weight, ADFI, ADG, and FCR data. There was no significant influence of the dietary supplements containing EGCG on the growth performance of the weaning rabbits (*p* > 0.05).

### 3.2. Antioxidant Status

The data presented in Table 3 and Table 4 demonstrate the effects of different doses of EGCG on the antioxidant status of the liver and jejunal mucosa in the rabbits. Dietary supplementation with EGCG had a significant influence on the MDA content, as well as the activity of the GSH-Px and T-AOC contents in both the liver and jejunal mucosa (*p* < 0.05). Specifically, the MDA contents in both the liver and jejunal mucosa significantly decreased with EGCG supplementation (*p* < 0.05). The linear and quadratic effects of EGCG on MDA content were significant, with an optimal dosage of 700 mg/kg and 878 mg/kg in the liver and jejunal mucosa, respectively. EGCG showed significant correlations with GHS-Px activity in the liver and jejunal mucosa (*p* < 0.05), with optimal doses of 1295 mg/kg and 765 mg/kg, respectively. Post-weaning rabbits fed a gradient concentration of EGCG exhibited significant (*p* < 0.05) linear and quadratic trends for T-AOC content in the liver (with the optimum EGCG dose being 750 mg/kg). Compared with the control group, the 600 and 1000 mg/kg groups significantly increased the T-AOC content in the jejunal mucosa. No significant influence of EGCG on the H_2_O_2_ content and the activities of the SOD in the liver and jejunal mucosa were observed (*p* > 0.05).

### 3.3. Immune Organ Index and Serum Immunoglobulin Content

The addition of EGCG to the diet resulted in a significant increase in the relative weights of the immune organs (Table 5) (*p* < 0.05). This study found that higher levels of EGCG in diets had a significant quadratic impact on the immune organ index, specifically for the spleen and thymus (*p* < 0.05). EGCG did not affect the development of the sacculus rotundus in the rabbits. The serum IgG content was significantly influenced by the EGCG, showing a quadratic relationship, and the optimal dosage was found to be 765.63 mg/kg (*p* < 0.05). There was no significant difference observed in the immunoglobulin levels of IgA and IgM among the groups (*p* > 0.05).

### 3.4. Intestinal Morphology

The results of the intestinal morphology are presented in Table 6. There were significant differences (*p* < 0.05) in the villus height and villus height/crypt depth ratio in the ileum across all the treatments. EGCG exhibited positive linear and quadratic effects on the villus height and villus height/crypt depth ratio of the ileum. Nevertheless, dietary supplementation with EGCG did not significantly affect the intestinal morphology of other intestinal segments in the post-weaning rabbits (*p* > 0.05).

### 3.5. Alpha Diversity Analysis

Table 7 displays the alpha diversity across all the groups. Dietary supplementation with EGCG exhibited a tendency to influence the Shannon index for cecal microbiota (*p* < 0.1). However, there was no significant effect of EGCG on the Chao1, dominance, pielou_e, and Simpson indexes among all the treatments (*p* > 0.05).

### 3.6. Cecal Microbial Compositions

Figure 1 illustrates the taxonomic distributions of the cecal microbiota among all the groups. At the phylum level, Firmicutes, Bacteroidota, and Verrucomicrobiota were dominant (>1%). The top five relative abundances at the phylum level in each treatment were Firmicutes, Bacteroidota, Verrucomicrobiota, Synergistota, and Desulfobacterota (Figure 1a). Figure 1b presents the top 10 family with Muribaculaceae and Clostridia_UCG-014 being the most prevalent, each accounting for 20% in total of all the groups. As shown in Figure 1c, the 600, 800, and 1000 mg/kg EGCG groups significantly increased the abundance of Euryarchaeota in comparison with the control treatment (*p* < 0.01). Additionally, the abundance of Patescibacteria was increased in rabbits treated with 200, 600, 800, and 1000 mg/kg EGCG. Dietary supplementation with 400, 800, and 1000 mg/kg EGCG significantly increased the abundance of Synergistota, while 800 mg/kg EGCG treatment significantly increased the *Desulfobacterota* abundance (Figure 1d, *p* < 0.01). A significantly reduced abundance of *Ruminococcus* was observed in the rabbits’ diets treated with EGCG, as compared to the control group (*p* < 0.05). The abundance of *Clostridia_vadinBB60_group* was extremely significantly decreased in the 600, 800, and 1000 mg/kg EGCG treatments compared with the control group (*p* < 0.01). Dietary supplementation with 800 mg/kg EGCG significantly increased the abundances of *NK4A214_group*, *Cloacibacillus*, and *dgA-11_gut_group*.

### 3.7. Beta Diversity Analysis

The PCoA and NMDS plots of the cecal microbiota were based on each weighted UniFrac Metric, which was quantified specifically using Adonis (*p* = 0.001, R^2^ = 0.291). According to the ANOSIM analysis (Appendix A), the cecal microbiota composition of the rabbits fed with 600, 800, and 1000 mg/kg EGCG deviated significantly from that of the control group (Figure 2).

Through the LEfSe analysis, there were 51 differentially abundant bacterial clades at the genus level (LDA score > 2.0) between all the treatments (Figure 3). *Methanobrevibacter*, *Monoglobus, Enterorhabdus*, *Sanguibacteroides*, and *Anaeroplasma*, were the dominant genera in the cecum when rabbits were fed with 600 mg/kg EGCG. In contrast, the cecum of rabbits treated with 800 mg/kg EGCG was inhabited mostly by *Cloacibacillus*, *Lawsonia*, *Candidatus_Saccharimonas*, *Subdoligranulum*, and *Anaerotruncus*. Treatment with 1000 mg/kg EGCG could enhance the abundances of *Synergistes*, *Coriobacteriaceas_UCG_002*, and *ADurbBin063_1*.

### 3.8. Correlation between Antioxidant Status and Gut Microbiota

A Spearman’s correlation analysis on the antioxidant capacity and microbiota was performed. As shown in Figure 4, positive and negative correlations were observed between various bacteria and the antioxidant capacity in the liver and jejunal mucosa. Extreme positive correlation was found between cecal ASV382 and ASV809 and liver MDA content. Liver GSH activity had an extreme negative correlation with ASV161, ASV234, ASV1024, and ASV1178, while a positive correlation with ASV6 and ASV98 was observed. There was a negative correlation between hepatic T-AOC and ASV351. The activity of GSH in the jejunal antioxidant system showed a positive correlation with ASV6 and a negative correlation with ASV9, ASV234, ASV351, and ASV1281, significantly.

### 3.9. Functional Predictions

According to the KEGG level 2 analysis, dietary supplementation with a high dosage of EGCG (800 and 1000 mg/kg) significantly enriched the enzymatic families, immune system, and metabolism (Appendix A). At KEGG level 3, 800 mg/kg EGCG treatment highly enriched transporters, bacterial chemotaxis, and bacterial motility proteins. The 1000 mg/kg EGCG group was highly enriched in its peptidases, starch and sucrose metabolism, and secretion system (Appendix A).

## 4. Discussion

Previous studies have demonstrated that tea polyphenols, including EGCG, play a crucial role in improving growth performance and modulating gut health in animals [20]. Hossain et al. [21] and Sarker et al. [22] demonstrated that green tea polyphenols positively affected weight gain in post-weaning mammals. Conversely, other researchers found no significant enhancement in pig- and broiler-fed diets supplemented with green tea polyphenols [23,24]. The present study found no statistical difference in the growth performance between the treatments supplemented with EGCG; however, EGCG numerically enhanced the 68-day body weight values compared with the control group. According to Zhang et al.’s [25] research, tea polyphenols can reduce body weight by enhancing metabolism, which is linked to their activity at the emulsion interface or interaction with enzymes that decrease weight gain. Under weaning stress, EGCG mitigated the associated damage; however, these protective actions were energy-intensive, leading to reduced energy availability for body weight gain [16,26]. In this study, rabbits fed with 400 mg/kg EGCG exhibited decreased weight during 68–88 d. This outcome was consistent with the observed enlargement of immune organs and the enhanced immune response, suggesting that energy was redirected toward immune function rather than growth. Consequently, no significant improvement in growth performance was found in this study.

Zhong et al. [27] reported that EGCG acts as a scavenger of various ROS/RNS. It is believed that EGCG mitigates the inflammatory response and oxidative stress, thereby reducing liver injury [28]. Our results demonstrated that the incorporation of EGCG into experimental diets positively influenced the antioxidant status in both the liver and jejunum. This observation is consistent with a previous study conducted by Al-Bajari et al. [26]. The antioxidant function of EGCG might be attributed to its ability to capture peroxyl radicals, thereby interrupting the chain reaction of free radicals and terminating lipid oxidation [27].

In the present study, we observed a reduction in MDA content in the EGCG-treated groups, indicating a potential decreased degree of lipid peroxidation. Ahmed et al. [29] reported that long-term treatment with a higher dosage of EGCG effectively reduced MDA formation in rats, which was similar to the present study. Cai et al. [30] reported that EGCG possessed an ortho-diphenoxyl functionality in its B ring, which could activate antioxidant activity. The present results showed that EGCG-treated weaning rabbits had the ability to improve their activity of GSH-Px both in the liver and jejunum. According to a study by Soussi et al. [31], the particular structure of polyphenols, including EGCG, could effectively reduce superoxide radicals and increase the expression of antioxidant enzymes such as catalase and GSH-Px. The antioxidant effects of tea polyphenols have been supported by numerous in vivo studies. Furthermore, our data corroborated previous observations [32,33], suggesting that EGCG can increase the content of antioxidants such as T-AOC, especially in liver tissues. These results clearly demonstrated that EGCG improved the antioxidant capacity and reduced the generation of peroxides through both enzyme-dependent and non-enzyme-dependent antioxidant pathways.

A previous study [34] corroborated our results, indicating that weaning stress could damage the immune system, thereby reducing the development of immune organs. Accumulating evidence has suggested that EGCG possesses immunomodulatory effects, with its impact on the immune response being dosage-dependent. In this study, diets supplemented with 400–500 mg/kg of EGCG resulted in the largest spleen and thymus sizes. Mice fed 0.15% and 0.3% EGCG diets exhibited no change while those fed 1% EGCG produced more proinflammatory cytokines [15]. This result reflected that high doses of EGCG might display toxicity to rats. As reported by Su et al. [35], EGCG might improve the thymus structure, accompanied with a higher thymus index and size. Our data also indicated that the supplementation of EGCG in the diets of post-weaning rabbits could increase their serum IgG levels, with an optimal dosage of 765 mg/kg. This result aligns closely with findings from rat studies [36]. Taken together, these studies underscore the necessity of determining and accurate dosage of EGCG, as an excessive amount might elicit an inflammatory response.

The intestine plays a central role in nutrient digestion and absorption, serving as a barrier to preventing the entry of exogenous harmful substances. Tang et al. [37] noted that reductions in VH and VH/CD reflected impaired intestinal mucosal function and diminished capacities for digestion and absorption capacities. Conversely, higher values of VH and VH/CD, along with a lower CD in the intestines, typically indicated better intestinal function. Bomba et al. [38] demonstrated that weaning stress resulted in lower VH and VH/CD in the ileum of piglets, a finding consistent with the present study. Thus, early weaning can lead to intestinal damage in mammals. The present study revealed that dietary supplementation with EGCG, particularly at 200 and 400 mg/kg, positively affected VH, CD, and VH/CD, corroborating findings from poultry research by Song et al. [16]. Furthermore, Hassan et al. [39] suggested that polyphenols possessed the ability to restore the intestinal surface. Taken together, these results indicate that dietary supplementation with EGCG suppresses oxidative stress and attenuates the damage to the immune system and mucosal barrier function caused by weaning stress in post-weaning rabbits.

A vast array of microbes inhabits the gastrointestinal tract, playing crucial roles in gut maturation and immune modulation. Stress induced by weaning often results in dysbiosis, characterized by an imbalance in gut microbiota, which increases the risk of pathogenic proliferation and diarrhea [40,41]. PCoA and ANOSIM analysis revealed distinct microbiota profiles in the rabbits treated with 600–1000 mg/kg EGCG compared to the control group. These changes in microbial diversity showed that EGCG exhibited greater selective antimicrobial activity, consistent with previous results [42]. Euryarchaeota are anerobic and lack defense mechanisms against oxidative stress [43]. As Murphy et al. [44] reported, *Desulfobacterota* serve various purposes, including protection against oxidation. The present study demonstrated that a higher dosage (600–1000 mg/kg) of EGCG in the diets of weaning rabbits resulted in the enrichment of Euryarchaeota and *Desulfobacterota*, potentially supporting the hypothesis that EGCG could alleviate oxidative damage. Lower enrichment with *Ruminococcus* and *Clostridia_vadin BB60_group* was observed in EGCG-treated groups, which could facilitate carbohydrate and amino acid metabolism and increase fat deposition [45,46,47]. The reduction in these bacteria could potentially explain the growth performance and nutrient digestibility (unlisted) without significant differences. Additionally, EGCG has been proven to maintain intestinal immune function by regulating the release of inflammation cytokines [48]. Functional predictions of microbiota indicated that diets supplemented with 800 and 1000 mg/kg EGCG could improve the immune system and protect the intestinal barrier. The immune response findings of this study indicated that the intermediate dosage of EGCG was associated with enhanced immune organ function. However, this observation presents a slight discrepancy with the microbial results and necessitates further investigations.

Both enzymatic and non-enzymatic systems play crucial roles in defending against oxidative stress. Among non-enzymatic antioxidants, short-chain fatty acids (SCFAs) produced by microbiota have been shown to effectively mitigate oxidative stress and repair intestinal barrier function [49]. The present study also supported this opinion, that alterations in the gut microbiota were linked to oxidative stress. The *Clostridium* cluster, a key SCFA-producing microbiota, plays an important role in maintaining the balance between oxidants and antioxidants. We found that Oscillospiraceae (ASV6)*,* the third Clostridiales family, was positively correlated with the antioxidant indexes. Moreover, the enrichment of Oscillospiraceae increased with EGCG supplementation, which was also identified as a healthy gut maker [50]. *Parabacteroides* (ASV351) has a close relationship with host health, which was increased and significantly positively correlated with obesity-related indexes in mouse studies [51]. We also found that its levels were decreased and negatively correlated with antioxidant capacity in the weaning rabbits fed with EGCG. Previous studies found that Rikenellaceae (ASV9) played a vital role in lipid metabolism [52]. *Elusimicrobium* (ASV1281) in the Elusimicrobiota phylum was decreased in the cecal samples of higher-dosage EGCG groups, which might have mediated amino acid metabolism [53]. The abundance of Rikenellaceae and Elusimicrobiota being negatively correlated with antioxidant capacity was confirmed in this study. The underlying mechanism is not clear and needs further exploration. Based on these results, we believe that gut microbiota, mediated by EGCG, could successfully alleviate the effects of oxidant stress.

## 5. Conclusions

The findings of this study indicated that diets supplemented with EGCG in post-weaning rabbits could alleviate oxidative stress induced by weaning, including enhanced antioxidant capacity, reduced peroxide generation, increased formation of antioxidant enzymes, improved intestinal morphology, as well as reduced intestinal inflammation. Additionally, the composition and function of the cecal microbiota changed with high-dose (600–1000 mg/kg) EGCG treatment. EGCG could elevate the relative abundances of Euryarchaeota, Desulfobacterota, and Oscillospiraceae, thereby supporting intestinal barrier and preserving antioxidant capacity. This study suggests that high dosages of EGCG (400–800 mg/kg) supplementation could help suppress oxidative stress and attenuate weaning stress, thereby contributing to improved post-weaning outcomes in rabbits.

## Figures and Tables

**Figure 1 animals-14-03011-f001:**
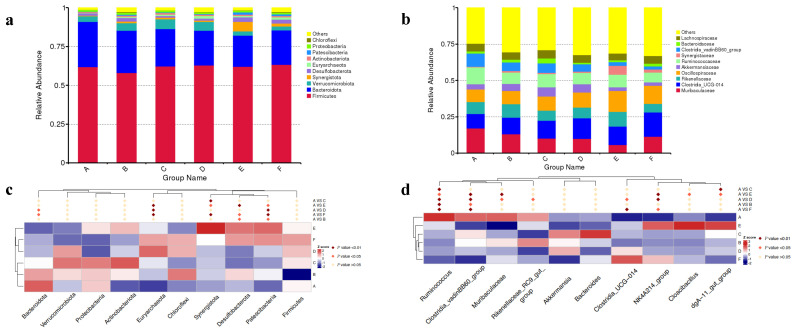
Relative top 10 abundances at the phylum (**a**) and family level (**b**) in the rabbit cecum treated with different dosages of EGCG. Significantly different microbiota between EGCG-treated groups and the control treatment at the (**c**) phylum level and (**d**) genus level. A The control group; B 200 mg/kg EGCG-treated group; C 400 mg/kg EGCG-treated group; D 600 mg/kg EGCG-treated group; E 800 mg/kg EGCG-treated group; F 1000 mg/kg EGCG-treated group.

**Figure 2 animals-14-03011-f002:**
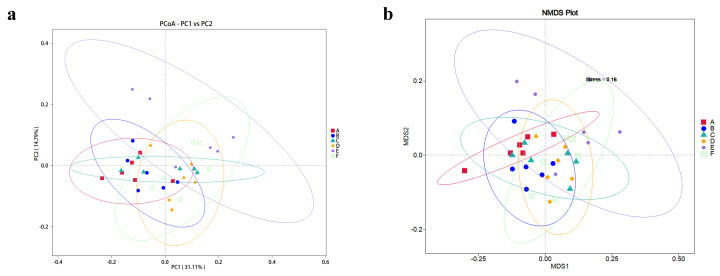
(**a**) PCoA plot and (**b**) NMDS plot of the beta diversity analysis of cecal samples from rabbits fed with EGCG. A The control group; B 200 mg/kg EGCG-treated group; C 400 mg/kg EGCG-treated group; D 600 mg/kg EGCG-treated group; E 800 mg/kg EGCG-treated group; F 1000 mg/kg EGCG-treated group.

**Figure 3 animals-14-03011-f003:**
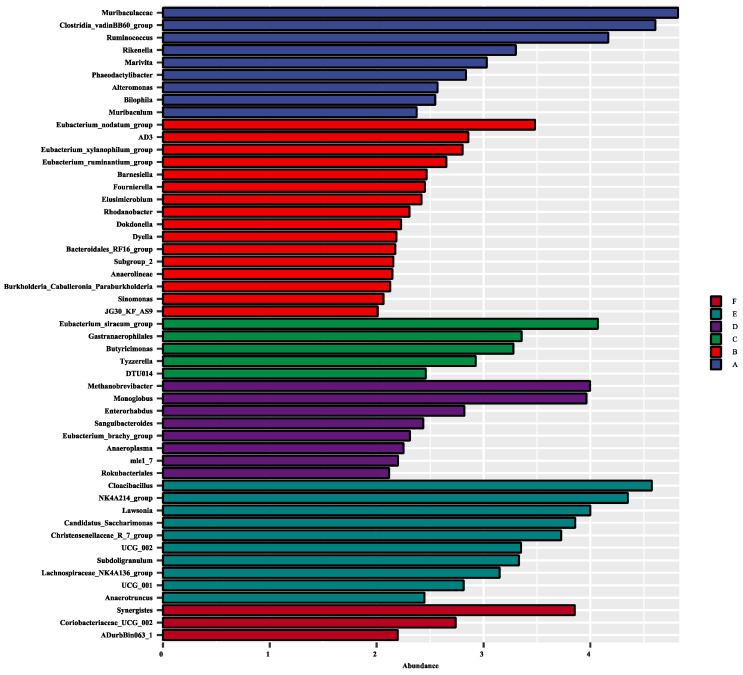
Effect size (LEfSe) analysis based on the genus level between treatments. A The control group; B 200 mg/kg EGCG-treated group; C 400 mg/kg EGCG-treated group; D 600 mg/kg EGCG-treated group; E 800 mg/kg EGCG-treated group; F 1000 mg/kg EGCG-treated group.

**Figure 4 animals-14-03011-f004:**
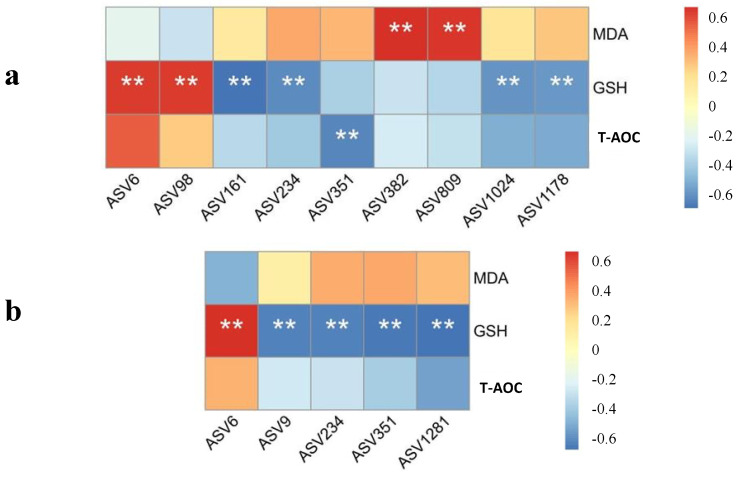
Spearman heatmap correlation between the antioxidant capacity in the (**a**) liver and (**b**) jejunal mucosa and the gut microbiota at the ASV level. ** presented an extremely significant difference (*p* < 0.01).

**Table 1 animals-14-03011-t001:** Feed ingredients and nutrient composition of basic diets.

Ingredients	%	Nutrient Level	%
Corn	24.00	DE/(MJ/kg)	10.12
Wheat bran	18.00	CP	15.77
DDGS	12.00	CF	16.49
Peanut wine	13.60	NDF	32.34
Peanut shell	20.00	ADF	21.40
Soybean meal	11.00	ADL	6.39
Bone meal	0.40	Lys	0.76
Premix ^(1)^	1.00	Sulfur amino acid ^(2)^	0.60
Total	100.00	Ca	0.61
		Total P	0.43

^(1)^ The premix provided the following per kg of diet: Fe 70 mg, Cu 20 mg, Zn 70 mg, Mn 10 mg, Se 0.25 mg, I 0.2 mg, VA 10 000 IU, VD 900 IU, VE 50 mg, VK 2 mg, D-pantothenic acid 20 mg, VB_12_ 0.02 mg, niacin 50 mg, Lys 2 g, Met 1.5 g. ^(2)^ The lysine and sulfur-containing amino acid were calculated values. The other nutrient values were measured values.

**Table 2 animals-14-03011-t002:** Influence of diet supplemented with EGCG on the growth performance of weaning rabbits.

	EGCG Level (mg/kg)							*p*-Value		
Item	Control	200	400	600	800	1000	SEM	ANOVA	Linear	Quadratic
Body weight, kg										
40 d	0.68	0.68	0.68	0.69	0.66	0.68	0.004	0.272	0.541	0.651
68 d	1.16	1.19	1.25	1.30	1.19	1.26	0.009	0.269	0.365	0.540
88 d	1.87	1.81	1.67	1.84	1.72	1.84	0.028	0.188	0.737	0.292
40–68 d										
ADFI, g	62.14	61.55	62.14	60.95	61.07	62.15	0.238	0.491	0.598	0.242
ADG, g	17.43	18.35	20.70	21.83	19.20	21.07	0.719	0.473	0.150	0.356
FCR	3.56	3.35	3.00	2.79	3.18	2.94	0.081	0.571	0.241	0.464
68–88 d										
ADFI, g	164.00	163.42	160.58	157.17	159.42	163.17	1.000	0.313	0.273	0.639
ADG, g	32.40	30.72	26.57	31.62	29.43	28.71	0.874	0.346	0.135	0.151
FCR	4.92	5.38	6.04	5.48	5.44	5.84	0.167	0.904	0.185	0.457
40–88 d										
ADFI, g	107.92	106.49	104.24	101.67	101.32	103.78	1.300	0.697	0.177	0.382
ADG, g	25.12	23.50	23.34	23.93	23.61	24.25	0.589	0.221	0.546	0.063
FCR	4.27	4.53	4.46	4.27	4.29	4.27	0.148	0.206	0.384	0.277

**Table 3 animals-14-03011-t003:** Influence of diet supplemented with EGCG on the liver antioxidant status of weaning rabbits.

	EGCG Level (mg/kg)							*p*-Value		
Item	Control	200	400	600	800	1000	SEM	ANOVA	Linear	Quadratic
MDA, nmol/mL	9.74 ^b^	6.35 ^a^	6.03 ^a^	5.28 ^a^	5.11 ^a^	6.57 ^a^	0.436	0.017	0.027	0.002
H_2_ O_2_, nmol/L	3.73	3.74	3.44	4.13	4.04	4.01	0.110	0.488	0.252	0.512
GSH-Px, U/mL	37.40 ^a^	42.30 ^ab^	37.87 ^a^	50.69 ^c^	53.41 ^c^	46.99 ^bc^	1.397	0.001	0.001	0.001
SOD, U/mL	35.08	35.82	41.68	39.00	35.19	35.12	0.992	0.273	0.855	0.167
T-AOC, U/mL	0.68 ^a^	0.68 ^a^	1.03 ^b^	0.93 ^b^	0.99 ^b^	0.98 ^b^	0.037	0.003	0.002	0.002

^a,b,c^ Values without common superscripts in the same row differ significantly (*p* < 0.05).

**Table 4 animals-14-03011-t004:** Influence of diet supplemented with EGCG on the antioxidant status in the jejunal mucosa of weaning rabbits.

	EGCG Level (mg/kg)							*p*-Value		
Item	Control	200	400	600	800	1000	SEM	ANOVA	Linear	Quadratic
MDA, nmol/mL	14.91 ^b^	9.17 ^a^	11.20 ^ab^	10.71 ^ab^	7.81 ^a^	9.53 ^a^	0.672	0.032	0.014	0.024
H_2_ O_2_, nmol/L	2.74	3.16	3.17	2.87	2.85	2.88	0.684	0.365	0.653	0.374
GSH-Px, U/mL	40.06 ^a^	49.73 ^b^	53.51 ^b^	54.57 ^bc^	60.58 ^c^	54.76 ^bc^	1.306	0.001	0.001	0.001
SOD, U/mL	39.83	36.37	37.24	39.07	43.60	38.81	1.117	0.541	0.484	0.754
T-AOC, U/mL	0.74 ^a^	0.77 ^ab^	0.98 ^ab^	1.06 ^b^	0.79 ^ab^	1.05 ^b^	0.042	0.054	0.047	0.103

^a,b,c^ Values without common superscripts in the same row differ significantly (*p* < 0.05).

**Table 5 animals-14-03011-t005:** Influence of diet supplemented with EGCG on the immune organ index and serum immunoglobulin levels of post-weaning rabbits.

	EGCG Level (mg/kg)							*p*-Value		
Item	Control	200	400	600	800	1000	SEM	ANOVA	Linear	Quadratic
Spleen, g/kg	0.51 ^a^	0.58 ^a^	0.79 ^ab^	0.59 ^a^	0.63 ^a^	0.97 ^b^	0.463	0.027	0.271	0.003
Thymus, g/kg	0.88 ^a^	1.70 ^c^	1.58 ^bc^	1.31 ^b^	1.03 ^ab^	1.56 ^bc^	0.091	0.002	0.118	0.002
Sacculus rotundus, g/kg	1.02	1.10	0.81	0.86	0.63	0.95	0.421	0.362	0.203	0.315
IgM, ng/mL	51.14	50.53	49.62	55.60	58.65	51.14	1.22	0.253	0.088	0.233
IgA, ng/mL	90.51	99.62	95.24	106.01	97.78	90.51	2.31	0.558	0.153	0.130
IgG, ng/mL	1161.86 ^a^	1300.29 ^ab^	1319.78 ^ab^	1405.28 ^b^	1451.27 ^b^	1161.86 ^a^	28.34	0.016	0.175	0.005

^a,b,c^ Values without common superscripts in the same row differ significantly (*p* < 0.05).

**Table 6 animals-14-03011-t006:** Influence of diet supplemented with EGCG on the intestinal morphology of weaning rabbits.

	EGCG Level (mg/kg)							*p*-Value		
Item	Control	200	400	600	800	1000	SEM	ANOVA	Linear	Quadratic
Duodenum										
VH, μm	823.33	822.00	618.21	874.65	611.03	786.41	38.870	0.215	0.591	0.644
CD, μm	166.61	100.99	139.84	130.35	122.96	103.14	12.040	0.658	0.254	0.523
VH/CD	7.08	8.43	5.68	7.40	5.13	8.14	0.530	0.409	0.794	0.680
Jejunum										
VH, μm	882.48	731.79	568.03	648.49	735.78	667.48	40.030	0.328	0.262	0.154
CD, μm	159.75	124.12	200.99	154.54	139.67	160.74	10.339	0.407	0.908	0.920
VH/CD	6.84	6.22	3.12	5.02	5.68	5.11	0.382	0.349	0.414	0.278
Ileum										
VH, μm	506.26 ^a^	800.11 ^b^	719.12 ^b^	467.76 ^a^	420.07 ^a^	421.41 ^a^	37.383	0.002	0.026	0.020
CD, μm	159.57	110.84	156.58	140.76	193.67	205.33	11.333	0.163	0.060	0.073
VH/CD	4.16 ^b^	4.59 ^b^	7.54 ^c^	3.37 ^ab^	2.24 ^a^	2.27 ^a^	0.491	0.001	0.001	0.001

^a,b^ Values without common superscripts in the same row differ significantly (*p* < 0.05).

**Table 7 animals-14-03011-t007:** Influence of diet supplemented with EGCG on the alpha diversity of the cecal microbiota of post-weaning rabbits.

	EGCG Level (mg/kg)							
Item	Control	200	400	600	800	1000	SEM	*p*-Value
Chao1	1201.99	1295.19	1250.21	1233.38	1087.81	1215.51	24.052	0.218
Dominance	0.017	0.008	0.008	0.006	0.016	0.006	0.0018	0.343
observed_otus	1199	1292.33	1247.5	1231.5	1086.33	1212.33	24.051	0.225
Pielou_e	0.835	0.856	0.855	0.863	0.819	0.855	0.0055	0.174
Shannon	8.543	8.839	8.796	8.853	8.253	8.753	0.0705	0.088
Simpson	0.983	0.992	0.992	0.994	0.984	0.994	0.0018	0.343

## Data Availability

The datasets presented in this study can be found in online repositories. The names of the repository/repositories and accession number(s) can be found at: https://www.ncbi.nlm.nih.gov/ (accessed on 12 September 2024) PRJNA1151226.

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
