# Peer review of "Assessment of the Impact of Dietary Supplementation with Epigallocatechin Gallate (EGCG) on Antioxidant Status, Immune Response, and Intestinal Microbiota in Post-Weaning Rabbits"

_animals, 2024, doi:10.3390/ani14203011_

Round 1
Reviewer 1 Report
Comments and Suggestions for Authors
This sentence ,,Thus, this study provides experimental evidence that supports the utilization of EGCG as a viable approach to mitigate post-weaning stress in rabbits.'' it could be a conclusion. Have to be deleted from the introduction (lines 81-82).
At ,,2.1. Materials'' you have to present more details about the product: form of presentation, etc
At ,,2.2. Experimental design and animal treatment'' you have to detail how you included the supplement into the rabbit diet.
Please mention, from which anatomical part you collected the blood? And in what type of recipients? You have to detail how the rabbits were sacrificed, where was this process done and if was respected the regulation in force. What have you done with the dead bodies after you collected the samples? All these details have to be presented in ,,Materials and methods’’.
At ,, 2.5.1. Antioxidant indicators’’ – what type of assay kits (line 139)? What equipment and method you used?
The 16s rRNA sequencing was done by the authors of the manuscript, or was done in a specialised laboratory? Please clarify this!
Figure 1 is too small and unreadable.
Author Response
Q1. This sentence, Thus, this study provides experimental evidence that supports the utilization of EGCG as a viable approach to mitigate post-weaning stress in rabbits.'' it could be a conclusion. Have to be deleted from the introduction (lines 81-82).
Answer: Thanks for your suggestion, we have revised this statement and signed in red.
Q2. At, 2.1. Materials'' you have to present more details about the product: form of presentation, etc
Answer: Thanks for your advice, we have provided detail information about EGCG in “Materials” part and signed in red.
Q3. At 2.2, Experimental design and animal treatment'' you have to detail how you included the supplement into the rabbit diet.
Please mention, from which anatomical part you collected the blood? And in what type of recipients? You have to detail how the rabbits were sacrificed, where was this process done and if was respected the regulation in force. What have you done with the dead bodies after you collected the samples? All these details have to be presented in ,,Materials and methods’’.
Answer: Thanks for your advices, all the ingredients in the basic diet and different dosage of EGCG were mixed together and then the feed was pelleted. Blood samples were collected from ear vein. In the materials and methods part, the sacrifice processes have been revised and expressed more clarity and detail which were signed in red.
Q4 At, 2.5.1. Antioxidant indicators’’ – what type of assay kits (line 139)? What equipment and method you used?
Answer: The kits numbers were added in the “material and methods” part, all the antioxidant indicators were detected by colorimetric method with a spectrophotometer.
Q5. The 16s rRNA sequencing was done by the authors of the manuscript, or was done in a specialised laboratory? Please clarify this!
Answer: The 16s rRNA sequencing and data analysis in this study were performed by the Novogene Company (Beijing, China). We have clarified this statement.
Q5.Figure 1 is too small and unreadable.
Answer: It’s our fault to make a wrong size of this figure. We have revised it and inserted a new one which can be seen clearly.
Reviewer 2 Report
Comments and Suggestions for Authors
In this article, the authors show the effect of epigallocatechin gallate (EGCG) supplementation on oxidative stress and weaning stress in rabbits. They suggest that high dose EGCG treatment could alleviate the effect of oxidative stress induced by weaning, including enhanced antioxidant capacity, improved intestinal morphology, reduced intestinal inflammation in rabbits. Although further research is required to understand the mechanism by which EGCG induces antioxidant effects, the results of this study provide useful information on how to improve weaning stress in rabbits.
The topic addressed is interesting, but I think there are some revisions that should be made before publication.
1) Line 96. Both female and male rabbits are used in this study, but are there any differences between males and females in each measurement? Is there any gender bias, especially in measurements such as blood tests where the number of rabbits is small?
2) Line 197-201. Please add information on the presence of symptoms, such as diarrhea, that may be caused by weaning stress.
3) Line 200-201. When you say improvement in the 68-day body weight, which number do you mean? Please explain it in more detail.
4) Table 3-6. Please add the explanation of the significant difference indication.
Author Response
Q1 Line 96. Both female and male rabbits are used in this study, but are there any differences between males and females in each measurement? Is there any gender bias, especially in measurements such as blood tests where the number of rabbits is small?
Answer: This is a good question. In this study, we selected half males and females. Males and females rabbits were kept in separate cages to test the growth performance. There is no gender bias during sample collection. Blood samples were collected from 6 rabbits of each treatment including 3 males and 3 females. The slaughter samples were done with the same process. Thus, we have considered the influence of gender and eliminated it.
Q2 Line 197-201. Please add information on the presence of symptoms, such as diarrhea, that may be caused by weaning stress.
Answer: Many thanks for your good suggestion. During our experimental period, we really found some diarrhea in groups. But we didn’t do statistic on this data. As previous study report, weaning-induced stress will lead to loss of appetite, post-weaning diarrhea, growth retardation, intestinal inflammation, oxidant stress and un-balanced gut microbiota (Muller et al., 2019; Ma et al., 2021). We focused on the oxidant stress, intestinal immune status and cecal microbiota under weaning stress. We also found that dietary supplementation with EGCG could alleviate these poor performance caused by weaning stress. As you said, diarrhea is a significant symptom. We will collect this data to make our research more scientific in the next experiment.
Q3 Line 200-201. When you say improvement in the 68-day body weight, which number do you mean? Please explain it in more detail.
Answer: In the 68d body weight, the body weight in control group was 1.16kg, the other treatments values were higher than that. However, there was no significant difference among the treatment, we have deleted this statement.
Q4 Table 3-6. Please add the explanation of the significant difference indication.
Answer: Thank you for your advice. The explanation of different superscripts were added below the tables.
Reviewer 3 Report
Comments and Suggestions for Authors
Manuscript animals-3239994, entitled “Assessment of the impact of dietary supplemented with Epigallocatechin Gallate (EGCG) on antioxidant status, immune response, and intestinal microbiota for post-weaning rabbits”
Recommendation: The above paper is not suitable for publication in its present form.
This article provides information on the effects of Epigallocatechin Gallate (EGCG) dietary supplementation on antioxidant status, immune response, and intestinal microbiota in post-weaning rabbits. It is in general appropriately organized, carried out and written, however there are some points that should be corrected or clarified.
L2: “supplementation” instead of “supplemented”
L4: “in” instead of “for”
L12: “levels” instead of “the formation”
L19: “main” instead of “mainly”
L26: “did not receive EGCG and served” instead of “received therapy”
L31: What about IgG?
L31: No linear (Table 5)
L33: “observed” instead of “seen”
L35: What about 200 mg/kg?
L42: “systems”
L47-48: “…which is associated with substantial stress, impairing body weight gain.”
L63-64: “…benefits [11]. A previous report states that the biological activity of EGCG is due to its active phenolic…”
L70-71: “…levels of 300, and 600mg/kg EGCG could…”
L73-74: Not necessary here. Please delete
L88: “…and female, with a mean weight of 500g, approximately…”
L91: “feed” instead of “food”
L96: Both female and male? Two and two?
L98: “provided with” instead of “given”
L100: Please provide reference for “NRC allowances”
L111: “The rabbits were individually weighted at the age of 40 days, 68 days, and 88 days old.” Please add model of scale and precision
L112: “carried out” instead of “done” and “ingested” instead of “had”
L113-115: “…leftovers were weighed and subtracted from the total amount provided to calculate feed intake. Average weight gain (WG), feed intake (FI), and feed conversion…”
L117: “collected” instead of “taken”. 3 males and females?
L119: “obtain” instead of “get”
L120: “On day 88, a rabbit from each replicate with an average weight was selected…”
L127: “collected” instead of “taken”
L129: “…rotundus were also dissected and…”
L134: “Liver tissue and intestinal mucosa were weighed with a precision of ±0.1 g. Then…” Please add model
L140: “…of Biological Technology (Beijing, China), according to the manufacturer’s guidelines.”
L152-153: What do you mean by “The calculated ratio of VH to CD was got by using earlier measurements”?
L198: Please delete “Despite”
L200-201: Please delete. Not even a trend
L210, 212, 215: How were these levels calculated?
L215: “…the 600 and 1000…”
L218: Please delete “among all the groups”
L225: “organs”
L226: No linear effect
L231: “among” instead of “between”
In Table 5, please check superscripts for “Thymus”. It is not possible the superscript “a” to be in 400 mg/kg and not in 1000 mg/kg
L235: Please delete the second “are”
L236: Also in crypt depth
L240: Please check VH in jejunum
Figure 1 is not clear, so I cannot check the results presented in L251-268
L256-257: “…600, 800 and 1000 mg/kg EGCG groups…”
L259: “…with doses of 200, 600, 800 and 1000 mg/kg EGCG…”
L260: Please delete “Administering”
L260-261: “…with 400, 800 and 1000 mg/kg EGCG…”
L261-262: “…of Synergistota, while 800 mg/kg EGCG significantly increased Desulfobacterota abundance (Figure 1D; P < 0.01).”
L265: “…decreased in 600, 800 and 1000…”
L266-267: “supplementation” instead of “supplemented”
L267: “increased”
L278-279: “…with 600, 800 and 1000 mg/kg…”
In Figure 4, please replace “Total” with “T-AOC”
L314-315: “According to KEGG level 2 analysis, dietary supplementation with high dosages of EGCG (800 and 1000 mg/kg) significantly enriched enzyme families…”
L317, 318: Please delete “in”
L325: “…broilers fed diets supplemented with green tea polyphenols…”
L327: “…EGCG numerically enhanced 68-day body…”
L328: Please delete “results”
L331-332: “actions” instead of “reactions”
L345: “…peroxidation, and the present study…”
L349: “…150 mg/kg in rats…”
L349: “Was similar to the present study”? In your study, you did not have a EGCG level of 150 mg/kg.
L360: “pathways” instead of “functions”
L361: “A previous study [33] confirms our results, which shown that weaning…”
L365: You did not have a EGCG level of 500 mg/kg
L365-366: Incorrect statement. The greatest levels for EGCG at 1000 mg/kg
L380: “…particularly at 200 and 400 mg/kg, positively…”
L403-404: “…with 800 and 1000 mg/kg EGCG…”
L427: “…with EGCG in post-weaning…”
L429: Please delete “the”
Comments on the Quality of English LanguageModerate editing of English language required.
Author Response
Thank you for your detailed suggestions on the revision of our article. We have carefully read the whole article according to your requirements, corrected the wrong expression, and modified the language expression.
Please see the attachment.

Reviewer 4 Report
Comments and Suggestions for Authors
Title: Assessment of the impact of dietary supplemented with Epigallocatechin Gallate (EGCG) on antioxidant status, immune response, and intestinal microbiota for post-weaning rabbits
The manuscript “Assessment of the impact of dietary supplemented with Epigallocatechin Gallate (EGCG) on antioxidant status, immune response, and intestinal microbiota for post-weaning rabbits” suggested that the addition of large doses of EGCG (400-800 mg/kg) can effectively suppress oxidative stress and alleviate weaning stress, thereby contributing to the protection of post-weaning rabbits. It is well written article with some interesting findings; however, there are some corrections before its acceptance for publication:
Line 9-15: In the summary part, authors should describe the work simply and concisely to the public, that would be easy to understand for the layman as well. Therefore, I invite the authors to rewrite the summary.
Line 18: to benefit? Make correction.
Line 16-21: This information should be in introduction part of the manuscript, rather than in the abstract portion and it is making the abstract laborious.
I would suggest to write the abstract portion. It must be a single paragraph of about 200 words maximum based on the guidelines of the journal. Therefore, I would suggest to omit the unnecessary paragraphs and make this part attractive to the readers.
Line 41: The introduction is starting from sentence “Due to growing economic and societal pressures….” Authors may start as “The early weaning of rabbits has become a standard practice in the rabbit breeding industry..” likewise.
Line 63: Please follow the guidelines of the journal to cite any reference.
Line 88: Equal proportion of male and female?
Line 89: Mention the name of city and country.
Line 119: Samples were quickly frozen with liquid nitrogen or just placed at -20 ℃?
Line 121: How the rabbits were sacrificed or slaughtered, mention the guidelines followed by the authors.
Line 162: Please mention the name of the city.
Line 336: How authors can conclude that “Macromolecule damage caused by free radicals significantly influences the pathophysiology of atherosclerosis, inflammation, and toxicity”. I would suggest the authors discuss the results based on the findings of the current study, rather than assuming.
Line 345-347: The results should not be repeated in the discussion part.
Overall, in the discussion part, the authors should focus more on why the different levels of EGCG affect the antioxidant status, immune response, and intestinal microbiota in post-weaning rabbits
Line 436: Authors should suggest some guidelines for future research, i.e., which aspect should be focused on for future research.
English grammar and sentence structure should be revised and corrected throughout the manuscript.
Comments on the Quality of English LanguageEnglish grammar and sentence structure should be revised and corrected throughout the manuscript.
Author Response
Q1 Line 9-15: In the summary part, authors should describe the work simply and concisely to the public, that would be easy to understand for the layman as well. Therefore, I invite the authors to rewrite the summary.
Answer: Many thanks for your suggestion. We have rewritten the simple summary which can be more readable.
Q2 Line 18: to benefit? Make correction. Line 16-21: This information should be in introduction part of the manuscript, rather than in the abstract portion and it is making the abstract laborious. I would suggest to write the abstract portion. It must be a single paragraph of about 200 words maximum based on the guidelines of the journal. Therefore, I would suggest to omit the unnecessary paragraphs and make this part attractive to the readers.
Answer: Thanks for your suggestion. Yes, our abstract should be more concise and recapitulative. We have revised the abstract paragraph. Now the total words in abstract are less than 200.
Q3 Line 41: The introduction is starting from sentence “Due to growing economic and societal pressures….” Authors may start as “The early weaning of rabbits has become a standard practice in the rabbit breeding industry..” likewise.
Answer: This sentence has been revised and signed in red.
Q4 Line 63: Please follow the guidelines of the journal to cite any reference.
Answer: The citation format has been revised.
Q5 Line 88: Equal proportion of male and female?
Answer: Yes, half male and half female, the numbers are equal.
Q6 Line 89: Mention the name of city and country.
Answer: This company locates in Shenyang, China.
Q7 Line 119: Samples were quickly frozen with liquid nitrogen or just placed at -20 ℃?
Answer: Blood samples were not frozen in liquid nitrogen. Fresh blood was collected from ear vein and then centrifuged to get serum. After serum collection, the serum sample was stored in -20℃ for further analysis.
Q8 Line 121: How the rabbits were sacrificed or slaughtered, mention the guidelines followed by the authors.
Answer: Thanks for your suggestion, the slaughter and dissection process have been revised and stated more clearly. All the process followed the protocols approved by the Shenyang Agricultural University Laboratory Animal Welfare and Ethical Center.
Q9 Line 162: Please mention the name of the city.
Answer: Our 16s rRNA analysis is provided by Novogene Company (Beijing, China), this information has been stated. The information has been added and signed in red.
Q10 Line 336: How authors can conclude that “Macromolecule damage caused by free radicals significantly influences the pathophysiology of atherosclerosis, inflammation, and toxicity”. I would suggest the authors discuss the results based on the findings of the current study, rather than assuming.
Answer: It is a good suggestion. As we said in the introduction, weaning stress in young mammals leads to notable oxidative stress and apoptosis in the body (Muller et al., 2019). Oxidative stress arises from reactive oxygen species (ROS), signifying an inequilibrium between the generation of free radicals and the antioxidant capacity of animals (Orengo et al., 2021). The present study found that all the antioxidant indexes were lower in the control group in comparison with the EGCG groups. From these results we can’t get this conclusion, this sentence has been deleted.
Q11 Line 345-347: The results should not be repeated in the discussion part.
Answer: Thanks for your advice, this sentence has been revised.
Q12 Overall, in the discussion part, the authors should focus more on why the different levels of EGCG affect the antioxidant status, immune response, and intestinal microbiota in post-weaning rabbits
Answer: Thanks for your suggestion, in rats and poultry studies, the impact of EGCG may depend on the dosage administered which is also similar with our result. According to antioxidant capacity, immune organs indexes and intestinal microbiota results, we found that there was a quadratic relationship between EGCG level and body response. In rats’ study, high doses of EGCG displayed in vitro toxicity to rat hepatocytes (Chakrawarti, et al., 2016). At the same time, mice fed 0.15% and 0.3% EGCG diet exhibited no change while those fed 1% EGCG diet produced more proinflammatory cytokines (Pae et al., 2012). These results suggested that high doses of EGCG could result in poor performance in animals. The present study selected 6 levels of EGCG, our object aimed at finding a better dosage for rabbits. We also revised our discussion as your suggestion and labeled in red in L363-370 and L379-381
Q13 Line 436: Authors should suggest some guidelines for future research, i.e., which aspect should be focused on for future research.
Answer: This is a good suggestion, the relationship between immune response and microbiota should be focused in the future. Some discrepancy results were found in this study about the immune organ and microbiota function. This part needs further investigation.
Round 2
Reviewer 4 Report
Comments and Suggestions for Authors
The manuscript is sufficiently improved based on the comments of the reviewer and may be accepted in its current form for the possible publication in Animals.